# Integrated Transcriptome and Molecular Docking to Identify the Hub Superimposed Attenuation Targets of Curcumin in Breast Cancer Cells

**DOI:** 10.3390/ijms241512479

**Published:** 2023-08-05

**Authors:** Rui Wang, Hao Yu, Peide Chen, Ting Yuan, Jing Zhang

**Affiliations:** 1Jilin Provincial Key Laboratory of Livestock and Poultry Feed and Feeding in the Northeastern Frigid Area, College of Animal Sciences, Jilin University, Changchun 130062, China; ruiwang18@mails.jlu.edu.cn (R.W.); yu_hao@jlu.edu.cn (H.Y.); chenpd22@mails.jlu.edu.cn (P.C.); 2Institute for Cardiovascular Regeneration, Centre for Molecular Medicine, Goethe University Frankfurt am Main, 60629 Frankfurt am Main, Germany; 3Department of Medicine, Cardiology, Goethe University Hospital, 60590 Frankfurt, Germany

**Keywords:** curcumin, breast cancer, reverse docking, transcriptome

## Abstract

Numerous in vitro and in vivo studies have shown that curcumin primarily activates apoptotic pathways in cancer cells and inhibits cancer progression by modulating various molecular targets. In this study, we utilized reverse docking servers to predict 444 human proteins that may potentially be targeted by curcumin. Then, high-throughput assays were conducted by using RNA-seq technology on curcumin-treated MCF-7 (human breast cancer ER (+)) and MDA-MB-231 (human breast cancer ER(-)/TNBC) cancer cell lines. Enrichment analysis identified seven and eight significantly down-regulated signaling pathways in these two cell lines, where the enriched genes were used to construct protein–protein interaction networks. From these networks, the MCODE algorithm screened out 42 hub targets, which are core genes of the RTK-(PI3K-AKT)/(MEK/ERK1/2) crosstalk network. Genetic alteration and expression patterns of hub targets of curcumin may be closely related to the overall pathogenesis and prognosis of breast cancer. MAPKAPK3, AKT3, CDK5, IGF1R, and MAPK11 are potential prognostic markers and therapeutic targets of curcumin in patients with triple-negative breast cancer. Molecular docking and transcriptomic results confirmed that curcumin can inhibit these high-scoring targets at the protein level. Additionally, these targets can act as self-feedback factors, relying on the cascading repressive effects in the network to limit their own transcription at the mRNA level. In conclusion, the integration of transcriptomic and molecular docking approaches enables the rapid identification of dual or multiple inhibitory targets of curcumin in breast cancer. Our study provides the potential elucidation of the anti-cancer mechanism of curcumin.

## 1. Introduction

Breast cancer (BC) currently is the most prevalent form of cancer globally, accounting for 12.5% of all newly reported annual cancer cases worldwide [1]. Its incidence is particularly pronounced in high-income countries [2]. In the United States, breast cancer ranks as the most frequently diagnosed cancer among women, excluding nonmelanoma skin cancer. Each year, it accounts for approximately 30% (or 1 in 3) of all newly diagnosed cancers in females. After lung cancer, it is the second leading cause of cancer-related mortality in women [3,4,5]. Notably, breast cancer has exhibited a surge in previously low-incidence nations over the past decade. In the year 2022, it is estimated that China and the USA will witness approximately 4,820,000 and 2,370,000 new cancer cases, along with 3,210,000 and 640,000 cancer-related deaths, respectively [6,7]. Projections made by *The Breast* journal indicate that by 2040, the burden of breast cancer will escalate, surpassing 3 million new cases per year (a 40% increase) and exceeding 1 million annual deaths (a 50% increase) [6,7]. Thus, the urgency for the development of effective and affordable candidate drugs for improved breast cancer treatment becomes paramount, both in the realm of research and clinical applications. Researchers have increasingly focused on natural sources for the development of anti-tumor drugs, due to their well-established efficacy and safety profile [8].

Curcumin, a hydrophobic polyphenol derived from turmeric, a traditional spice in Indian cuisine, has garnered significant attention due to its medicinal properties in both Indian and Chinese medical systems [9]. Extensive research has been conducted on curcumin, with a focus on its anti-inflammatory, anti-angiogenic, antioxidant, wound healing, and anti-cancer effects [10,11,12]. Recent studies have demonstrated the multifaceted anti-cancer roles of curcumin across various types of cancer. Curcumin is able to impede cell proliferation and metastasis while also inducing cell death [13]. Considerable evidence substantiates the wide-ranging potential application of curcumin in the treatment of breast cancer. Curcumin exerts its anti-breast cancer effects by targeting various regulatory proteins, such as kinases, transcription factors, receptors, enzymes, growth factors, molecules associated with the cell cycle and apoptosis, and microRNAs. Additionally, curcumin has demonstrated its capacity to modulate several pivotal signaling pathways implicated in breast cancer progression and development, such as JAK/STAT, NF-κB, WNT/β-catenin, PI3K/Akt/mTOR, MAPK, apoptosis, and cell cycle pathways [14,15,16,17,18]. It is worth noting that curcumin exhibits direct interactions with over 30 different proteins, including DNA polymerase, focal adhesion kinase (FAK), thioredoxin reductase, protein kinase C (PKC), lipoxygenase (LOX), and tubulin. This further elucidates its mechanistic involvement in various cellular processes [19,20,21,22,23,24,25,26].

In recent decades, structure-based virtual screening for the identification of potential compounds has been used to design anti-breast cancer drugs [27]. Moreover, an inverse approach to virtual screening has gained traction, wherein the identification of targets for compounds within protein structure databases is conducted with high throughput. This approach serves to streamline the drug discovery process and minimize both the time and financial resources expended during target identification. The main purpose of our study is to propose a novel strategy that utilizes high-throughput technologies to screen for reliable down-regulated curcumin docking targets in breast cancer at the transcriptome level using RNA-seq. This integrated approach holds promise for unraveling the intricate interactions between curcumin and its target molecules, ultimately facilitating the identification and elucidation of novel therapeutic targets.

## 2. Results

### 2.1. Reverse Virtual Screening of Potential Targets of Curcumin

To predict the interactions between proteins and drug-like molecules, three representative reverse docking servers, Galaxy Sagittarius (structure- and similarity-based prediction) [28], PharMmaper (Pharmacophore-based) [29], and SuperPred (a combination of 2D, 3D, and fragment similarity values) were used in the study [30]. Based on the three computational approaches, a total of 444 candidate targets were identified (Appendix A). The Venn diagram showed that the targets predicted by the three servers have less than a 15% overlap with each other (Figure 1A). Then, the scatter plots were plotted for each set of targets on two representative breast cancer cell lines (MDA-MB-231 and MCF-7) based on the normalized gene counts provided by CCLE. Each point on the plot was assigned a different color to represent its the respective docking score. Then, the expression of targets was also compared. As shown in Figure 1B–D, the expression levels of the majority of targets remained consistent across the two cell lines in all three groups, with data points scattered around the regression line, whereas the points that are distributed differently (outside the region of the regression line) belong to specific targets of cell lines, such as EGFR, ESR1, etc.

### 2.2. Effects of Curcumin on the Transcriptome Profiles of Breast Cancer Cells

To investigate the inhibitory effect of curcumin on the proliferation of breast cancer cells, MCF-7 and MDA-MB-231 cells were cultured and treated with varying concentrations of curcumin for 24 and 48 h. The results showed that curcumin dose-dependently inhibited cell survival, as depicted in Figure 2A,B. In contrast, curcumin exhibited a much lower cytotoxic effect on MCF-10a cells, which are a part of a normal human mammary epithelial cell line. The IC50 value for MCF-10a cells was recorded as 118.60 μg/mL, which is more than twice the IC50 values of MCF-7 and MDA-MB-231, as shown in Appendix A. Subsequently, the IC50 value of curcumin at 48 h was selected to treat MCF-7 and MDA-MB-231 cells for transcriptome sequencing. The results of the differential expression analysis revealed a total of 2740 up-regulated transcripts and 3893 down-regulated transcripts in the MCF-7 cell line. Similarly, the MDA-MB-231 cell line exhibited 4619 up-regulated transcripts and 1964 down-regulated transcripts. In addition, we colorized the curcumin targets on the volcano plots according to the initial expression (TPM) before treatment. We can see that the expression levels of most of the targets are relatively low (blue), and the fold changes are less than 2 (Figure 2C,D). Moreover, cancer pathway (KEGG) enrichment analysis was performed for down-regulated differentially expressed genes (DEGs), and a tornado diagram is presented to compare the significant enriched pathways between the two cell lines, as shown in Figure 2E.

### 2.3. Construction of the PPI Network and Module Analysis

To better define the molecular mechanism of curcumin in breast cancer, the down-regulated DEGs in significantly enriched pathways and all down-regulated targets of curcumin were selected for protein interaction network analysis. Considering that curcumin can regulate downstream genes through transcription factors, we constructed a drug–transcription factor–gene interaction network based on the TRRUST database. The results showed that in the dataset of MCF-7 cells, curcumin targeted 10 transcription factors, which regulated 27 genes in the significantly down-regulated pathway. In contrast, the dataset of MDA-MB-231 cells revealed that curcumin only targeted two transcription factors (STAT1 and NR3C2), both of which regulated the same downstream gene, EGFR. Notably, EGFR is also a direct target of curcumin (Figure 3A,B). Furthermore, two PPI networks were constructed separately using Metascape. Each network contained 484 and 345 interacting proteins. The MCODE algorithm was further used to identify the tightly linked network components in both networks. In total, 7 and 4 mcodes were identified from them, each containing 25 and 26 curcumin targets. We found that curcumin targets in these modules exhibited high centrality in the original PPI network. We inferred that the binding of curcumin and these hub targets can initiate a cascade of repressive effects, leading to the significant inhibition of tumor-associated pathways and ultimately attenuating their own transcription (Figure 3C,D).

### 2.4. Screening and Docking Evaluation of Effective Superimposed Attenuation Targets

Next, we redocked the 42 central targets provided by MOCDE with curcumin by using CB-Dock2 and integrated the docking score with the initial abundance and down-regulation ratio of the targets in the two cell lines, as shown in Figure 4A,B. All hub targets are listed in Figure 4A,B, and we found that some targets had mRNAs expressed at low levels or with relatively low down-regulation folds, indicating their weak ability to recruit curcumin. We selected 10 genes (MAPK3, CDK4, RXRA, AKT1, IGF1R, RARA, ESR1, STAT1, MAP2K2, and NFKB1) from the MCF7 cell line and 14 genes (PTPN11, MAPK3, MAPKAPK3, ITGB1, MAPK12, MAPK14, AKT3, CDK5, CDK6, KIT, STAT1, IGF1R, MAPK11, and EGFR) from the MDA-MB-231 cell line as efficient targets of curcumin. These genes are able to attract more curcumin to attenuate the regulatory network and ultimately down-regulate their own mRNA expression levels. Based on the breast cancer signaling pathway map, we can observe that curcumin has the ability to exert a wide range of inhibitory effects as a triple target of the RTK-PI3K/AKT-ERK1/2 axis in luminal A/B, HER+ positive, and basal-like/TNBC breast cancers (Figure 4C). In addition, the molecular docking results of several infrequently studied core targets (PTPN11, MAPK3, IGF1R, STAT1, ESR1, and EGFR) with curcumin are presented in Figure 4D–K. The scores with the inhibitors provided by the template at the same binding site were also compared (Figure 4D–K).

### 2.5. Genetic Alteration and Prognostic Value of Curcumin Targets in Patients with Breast Cancer

To evaluate potential correlations between genetic alterations in 21 candidate targets of curcumin and prognosis in breast cancer, the cBioPortal tool was utilized in the study. A total of 1093 patients from the breast invasive dataset (TCGA, Firehose Legacy) were analyzed. Genetic alterations of each curcumin target mainly involved gene amplification and up-regulation of mRNA expression, with percentages ranging from 4% to 20% (Figure 5A,B). Among the five different types of breast cancer, the rate of gene alteration ranged from 62.14% to 92.86% (Figure 5C). Kaplan–Meier plots demonstrated that the genetic alteration of overall curcumin targets tended to be associated with inferior overall survival (logrank test *p*-value = 0.058) (Figure 5D). Together, the results suggest that genetic alterations in curcumin targets occur at a high rate in breast cancer patients and are associated with an unfavorable prognosis.

Furthermore, we investigated the prognostic value of each of the two groups of curcumin targets in their respective breast cancer subtypes (patients) by using the Kaplan–Meier plotter platform. The results showed that only five genes (MAPKAPK3, AKT3, CDK5, IGF1R, and MAPK11) were associated with lower progression-free survival (PFS) in the TNBC patient dataset. Among these genes, IGF1R exhibited the strongest significant correlation with prognosis in breast cancer patients (Figure 5E). Our analysis indicated that the transcriptional expression levels of MAPKAPK3, AKT3, CDK5, IGF1R, and MAPK11 represent prognostic factors for TNBC. These genes could potentially be used as biomarkers for evaluating prognosis and tailoring individualized therapy treatment with curcumin.

## 3. Discussion

It is well known that once the pathogenic genes of a disease are identified, computer-assisted screening and design of lead compounds for these genes or protein products can lead to the rapid development of new drugs. Under this incentive, the reverse docking method has attracted more and more attention in the search for unknown targets of natural products and existing drugs. A structure-based drug discovery strategy can search for a large number of unknown and unexpected proteins that may inadvertently bind to lead compounds, resulting in serious side effects. However, the number of high-scoring proteins provided by the molecular docking platform (dry experiment) is still too large for in vivo and in vitro verification (wet test) undertaken in the laboratory. Additionally, there is no high-throughput technology available to assist with this verification requirement. Currently, the widespread popularity of RNA-seq technology and the decreasing cost of sequencing provide an opportunity for data mining to verify reverse docking results. Therefore, the purpose of this study is to implement a novel research strategy by utilizing RNA-seq technology to assess the efficacy of targets. The targeted screening of curcumin in breast cancer in our study demonstrates the feasibility of this approach.

As a heterogeneous disease, breast cancer is composed of various molecular subtypes with different clinical and pathological features. The most commonly recognized breast cancer subtypes include luminal A, luminal B, HER2-enriched, and basal-like breast cancer (also known as triple-negative breast cancer). Each subtype is characterized by unique molecular markers and exhibits different prognoses and responses to treatments. When conducting in vitro experiments with breast cancer cell lines, the commonly used models are MCF-7 and MDA-MB-231. MCF-7 cells are representative of the luminal A subtype and exhibit hormone receptor expression, making them suitable for studying hormone-responsive breast cancers. MDA-MB-231 cells, on the other hand, belong to the basal-like subtype and lack expression of hormone receptors, thus representing the TNBC category. To date, the anti-cancer mechanism of curcumin has not been comprehensively elucidated due to its wide range of targets and involvement in multiple regulatory pathways. Fortunately, according to the in vitro experiments and transcriptome analysis (Figure 4C), we can infer that the pathways it depends on vary depending on the type of breast cancer. 

We initially focused on curcumin targets identified in MCF-7 cell lines, and in particular estrogen receptor-alpha (ERα), which is encoded by the ESR1 gene. This receptor experienced the greatest down-regulation (16-fold) in MCF-7 cells after curcumin treatment. Additionally, the docking result in Figure 4D demonstrated that ERα is a high-affinity target of curcumin. ERα is the driver of progression in most diagnosed breast cancers. Approximately 70–80% of breast cancer patients express high levels of ESR1 in tumor tissue, making it an important target for the treatment of this specific subtype of breast cancer. In the nucleus, the sex hormone 17β-estradiol (E2)–ER-α complex regulates gene transcription in conjunction with various coactivators and corepressors. Outside the nucleus, E2 is involved in activating multiple signaling pathways [31]. According to our predicted and subsequent validation, based on the CCLE expression profile data, the ESR1 gene was indeed observed to be almost completely repressed at the transcriptional level. We hypothesize that curcumin down-regulates the repression of the transcription factors (ESR1, STAT1, RXRA, RARA, and NFKB1), leading to the down-regulation of numerous downstream factors. This indirect effect ultimately impacts the expression of curcumin itself. Michael Sun et al. found that earlier studies have shown that curcumin inhibits the proliferation of human pancreatic cancer cells and down-regulates the ESR1 gene at the mRNA level. They found that the up-regulation of miR-22 may silence ESR1. To test this hypothesis, they transfected mimics of miR-22 [32]. Subsequently, Deo Prakash Pandey et al. also confirmed with the MCF-7 cell line that miR-22 could inhibit estrogen signaling by directly targeting estrogen receptor-α mRNA [33].

However, we found that curcumin targets HER2-enriched and basal-like breast cancer, which occupy a key node in a complex network of tumorigenesis. This network mainly consists of the PI3K/AKT and Ras/MAPK (Ras/Raf/MAPK (MEK)/ERK) signaling pathways (Figure 4B,C,E). Both the PI3K/AKT and Ras/MEK/ERK pathways are important intracellular signaling cascades that regulate cell growth, proliferation, and apoptosis, as well as invasion [34,35,36]. The oncogenic amplification and/or mutation of effectors in the PI3K/AKT and MAPK/ERK pathways frequently occur during tumorigenesis, resulting in the abnormal activation of signaling [34,37,38,39,40,41,42,43]. The PI3K/AKT and MAPK/ERK pathways are dysregulated in almost one-third of human cancers. In many cancers, these pathways are activated simultaneously, making them popular targets for cancer therapy [44,45]. The initial perspective on growth factor, hormone, and cytokine receptor signaling networks is that PI3K/Akt and MAPK/ERK are two distinct parallel pathways. In recent years, more studies have found that there are multiple intersections between these two pathways, and their synergistic effects determine cell fate. There are 802 interacting proteins involved in PI3K-mediated signaling [46], and over 2000 interactions associated with MAPK family kinases [47]. At least 284 of these interactions are components of the endogenous ERK1 complex [48]. Both the PI3K/AKT and MAPK/ERK pathways are activated by tyrosine kinase receptors (RTKs) or G-protein-coupled receptors (GPCRs) [35]. Since both the PI3K/AKT and MAPK/ERK pathways have significant functional overlap, the compensatory activation of one pathway will largely attenuate the effect of targeted inhibition on the other, leading to drug resistance [49]. Eung-Ryoung Lee et al. found that inhibiting AKT activation resulted in a significant increase in ERK1/2 phosphorylation. Chia-Hung Chen et al. confirmed that MEK inhibitors can induce AKT activation and drug resistance by inhibiting the negative feedback of ERK-mediated HER2 phosphorylation in breast cancer cell lines [50,51]. Therefore, the development of dual inhibitors targeting both AKT and ERK may address this disadvantage. According to the results of our molecular docking and transcriptome analysis, we have discovered that curcumin not only acts as a dual inhibitor of the PI3K/AKT/mTOR and RAF/MEK/ERK pathways in breast cancer cell lines, but it can also function as an inhibitor of multiple RTKs to deactivate downstream signaling pathways. Kyu Sic You et al. demonstrated that the combined inhibition of AKT and MEK pathways enhances the anti-cancer efficacy of EGFR-targeted gefitinib on TNBC cells [52]. By comparing the initial expression abundance and down-regulation fold of MAPK3 and AKT1, as well as AKT3 in the two cell lines, we are led to believe that curcumin primarily functions as an inhibitor of MAPK3 (ERK1) and inhibits AKT1 and AKT3 to block the compensatory effect of the PI3K-AKT pathway in MCF-7 and MDA-MB-231 cells, respectively (Figure 4A,B). In addition, our results also show that other targets (SRC, NFKB, and STAT1) are also involved in crosstalk in the above pathways, which has been widely reported [22,53,54,55,56,57,58,59]. Finally, in Figure 6, we demonstrate that curcumin could serve as an inhibitor for multiple targets of the RTK-(PI3K-AKT)/(MEK/ERK1/2) crosstalk network. We also provide a schematic diagram of blocking the compensatory action between the two pathways.

In conclusion, the curcumin reduced the mRNA expression levels of predicted targets in the breast cancer cells in the study. The curcumin not only blocked the downstream targets’ activity at the protein level, but also comprehensively repressed the tumorigenesis-promoting phosphorylation dominated by the RTK-(PI3K-AKT)/(MEK/ERK1/2) crosstalk network, thereby indirectly limiting their own transcription. Our study presents a novel high-throughput data integration analysis pipeline that combines transcriptomics technology with molecular docking technology. This pipeline enables the discovery of drug targets for superposition inhibition. The central role of these self-feedback targets in pharmacogenetic networks ensures the credibility of curcumin in breast cancer treatment. Our research strategy also reduces redundant candidate genes for later in vivo and in vitro validation and accelerates the elucidation and clinical application of the mechanism of action of new drugs.

## 4. Materials and Methods

### 4.1. Cell Culture

The MCF-7 (ER-positive), MDA-MB-231 (triple-negative) breast cancer cell line, and MCF-10A human mammary epithelial cell line were obtained from the American Type Culture Collection (ATCC). The MCF-7 cell line was cultured in Dulbecco’s Modified Eagle’s High-Glucose Medium supplemented with 10% FBS, 100 UI/mL penicillin, and 100 μg/mL streptomycin. MDA-MB-231 was cultured in RPMI 1640 medium supplemented with 10% FBS, 100 UI/mL penicillin, 100 μg/mL streptomycin, and 2 mM glutamine. MCF-10A was cultured in Dulbecco’s Modified Eagle’s Medium/Ham’s F12 Medium (1:1) supplemented with 5% horse serum, 20 ng/mL epidermal growth factor, 0.01 mg/mL insulin, and 500 ng/mL hydrocortisone. The cells were cultured in a humidified incubator at 37 °C and 5% CO_2_. 

### 4.2. Treatment Methods for Cells

Curcumin was purchased from Sigma-Aldrich Corporation (St. Louis, MO, USA), and was dissolved in DMSO at a concentration of 50 mM for storage and diluted to specific concentrations in the cell culture medium for cell treatments. The final concentration of DMSO in the above prepared treatment solutions was less than 0.1% (*v*/*v*). During treatment, the same volume of DMSO was added to the control groups. The curcumin stock solution was diluted in the cell culture medium to concentrations of 5, 10, 20, 40, 60, 80, 100, 120, and140 μM, and the same volume of DMSO was added to the control groups. When the confluence of breast cancer cells in the culture dish reached more than 80%, cell viability was tested after 48 h of treatment with the above concentrations. The IC20 and IC50 values were calculated by GraphPad Prism 9.0.

### 4.3. Cell Viability Assay

Cell viability was assessed by employing a cell counting kit (CCK-8; Dojindo, Tokyo, Japan). Breast cancer cells were seeded in 96-well plates at a density of 0.2 × 104 cells per well. After 24 h and 48 h of treatment with curcumin, 10 μL of CCK-8 reagent was added to each well. The plates were then incubated at 37 °C for 1 h, which was followed by measuring the absorbance value (OD) at 450 nm using a spectrophotometer, in accordance with the manufacturer’s instructions.

### 4.4. RNA-Seq and Data Analysis

MCF-7 cells were exposed to either the vehicle or a 40 μM concentration of curcumin for 48 h, while MCF-MB-231 cells were treated with either the vehicle or a 50 μM concentration of curcumin for the same duration. Following the treatment, the breast cancer cells were harvested and washed twice with cold PBS. Subsequently, total RNA was extracted and suspended in diethyl pyrocarbonate. The integrity and purity of the RNA were assessed, and qualified RNA samples were utilized for PCR amplification in order to construct a cDNA library. The cDNA library was sequenced using an Illumina HiSeq^TM^ 300 platform (Illumina, San Diego, CA, USA). The resulting sequencing reads were aligned utilizing the spliced read aligner HISAT2 [60], with the Ensembl human genome assembly (Genome Reference Consortium GRCh38) employed as the reference genome. DESeq2 [61] was used to identify differentially expressed genes (DEGs) separately. Genes exhibiting a false discovery rate (FDR) q-value of less than 0.05 and an absolute Log2 (fold change) greater than 1 were considered to be DEGs. The cancer pathway enrichment analysis was used to screen for the significant down-regulated DEGs of RNA-seq data in two breast cancer cell lines by using WebGestalt [62], protein–protein interactive (PPI) network analysis was performed using Metascape [63], and molecular complex detection (MCODE) clustering analysis was performed to detect the hub targets of curcumin in the PPI network [64]. All of the networks were visualized with Cytoscape 3.9.0 [65]. The Venn diagrams and heatmaps were constructed with TBtools v 1.120 [66]. The integrated scatter plots were generated by Oranges 3.35 (https://orangedatamining.com/docs/bioinformatics, accessed on 23 April 2023).

### 4.5. High-Throughput Target Prediction and Molecular Docking

Network pharmacology and molecular docking were employed to explore the mechanism of the anti-cancer effect of curcumin. The 3D structure of curcumin (ligand) was searched for on the pubchem website [67]. The 3D structure of proteins (receptor) were obtained from the Protein DataBank (https://www.rcsb.org/, accessed on 12 May 2023). Potential targets of curcumin were obtained from GalaxySagittarius [28], SuperPred [30], and PharmMapper [29]. Blind ligand–protein docking was conducted for curcumin against the structure of each protein using the CB-Dock2 web server (https://cadd.labshare.cn/cb-dock2/php/blinddock.php, accessed on 12 May 2023), as CB-Dock2 predicts cavities of the protein and calculates the centers and sizes of the top N (*n* = 5 by default) cavities [68].

### 4.6. cBioPortal Analysis and Kaplan–Meier Plotter

The cBio Cancer Genomics Portal (cBioPortal, https://www.cbioportal.org/, accessed on 16 July 2023) is an open access resource based on TCGA that provides multidimensional visual data. The type and frequency of 21 hub superimposed attenuation targets in breast cancer patients were analyzed using the “OncoPrint” and “Cancer Type Summary” modules. The genomic profiles included copy-number variance (CNV) from the breast invasive dataset (TCGA, Firehose Legacy), and mRNA expression z-scores (RNA Seq V2 RSEM). OS was calculated according to the cBioPortal’s online instructions. The prognostic value of each target of curcumin mRNA expression in breast cancer was assessed according to relapse-free survival (RFS) using the Kaplan–Meier plotter (kmplot.com/analysis, accessed on 16 July 2023), an online database that includes gene expression data and clinical data. With the purpose of assessing the prognostic value of a specific gene, the patient samples were divided into two cohorts by auto-selecting the best cutoff. Log-rank *p*-values and HRs with 95% confidence intervals were determined using this webpage.

## Figures and Tables

**Figure 1 ijms-24-12479-f001:**
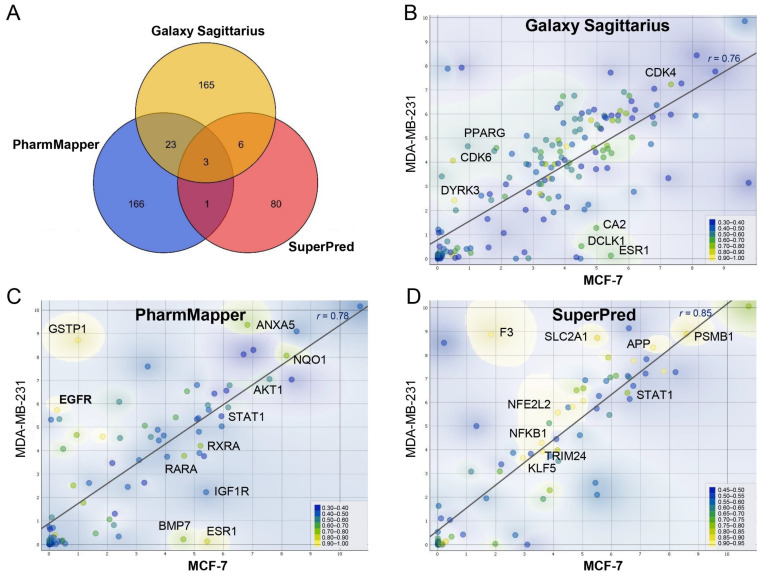
Reverse virtual screening of potential targets for curcumin. (**A**) The overlapping information of curcumin target prediction using three reverse docking systems; (**B**–**D**) comparison of the expression and docking scores of targets predicted by three reverse docking platforms in MCF-7 and MDA-MB-231 cell lines. Each data point in the scatter plot is mapped to the color scale, which varies with the docking score from the server.

**Figure 2 ijms-24-12479-f002:**
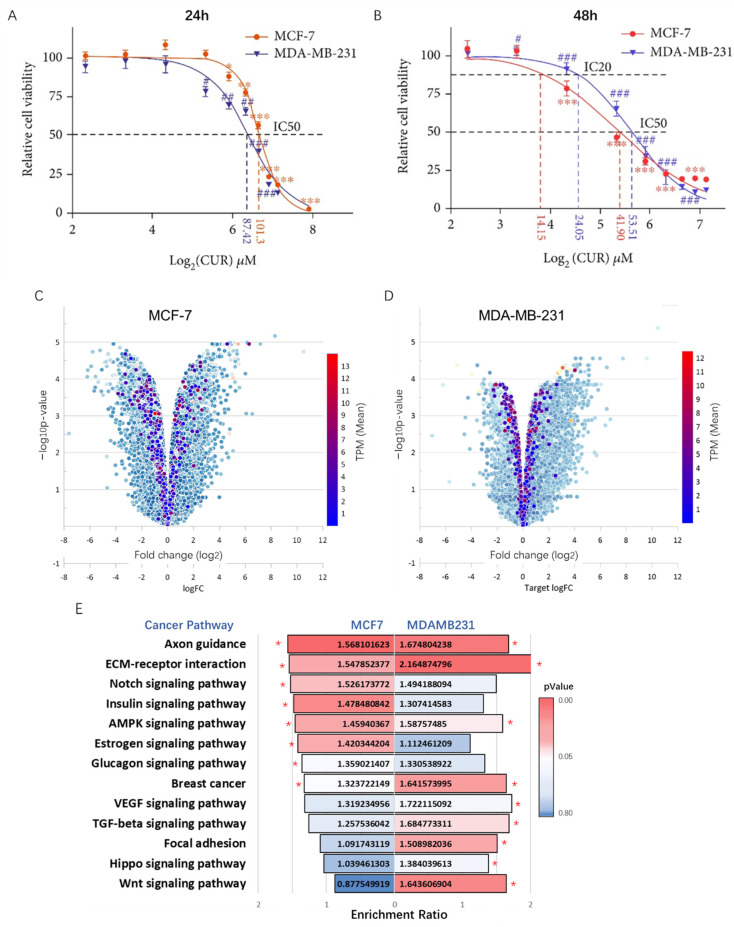
Effects of curcumin on the transcriptome profiles of breast cancer cells. (**A**,**B**) Evaluation of curcumin cytotoxicity on MCF-7 and MDA-MB-231 breast cancer cell lines after treatment with varying concentrations for different time intervals. Data are presented as mean (%) of control ± SE, *n* = 3. */# *p* < 0.05; **/## *p* < 0.01; ***/### *p* < 0.001. (**C**,**D**) Volcano plot showing the distribution of all differentially expressed transcripts in MCF-7 and MDA-MB-231 cell lines after curcumin treatment, with curcumin targets colored based on their initial expression levels (TPM). (**E**) Tornado diagram of significant enriched cancer-related pathways from the down-regulated DEGs in MCF-7 and MDA-MB-231 cells after treatment with curcumin, * *p* < 0.05.

**Figure 3 ijms-24-12479-f003:**
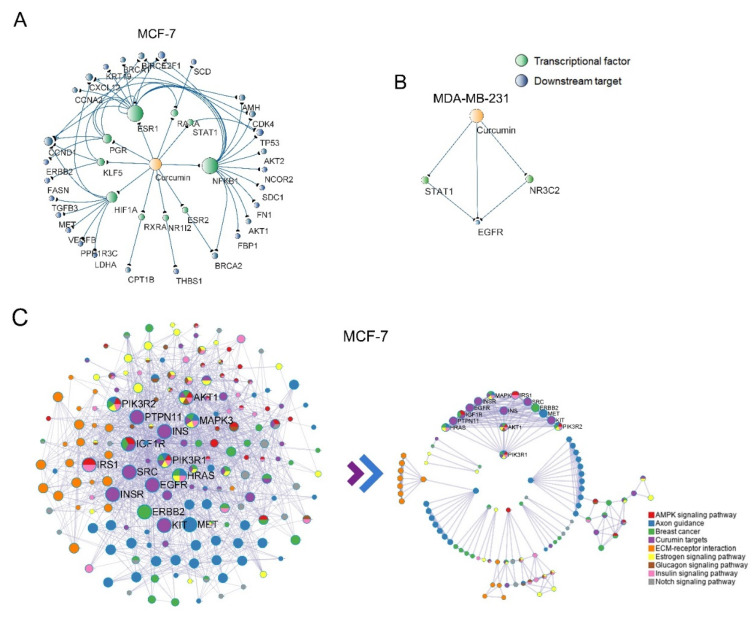
Construction of the PPI network and module analysis. (**A**,**B**) The drug–TF–gene interaction networks of curcumin in the significantly down-regulated pathways in the MCF-7 and MDA-MB-231 cells; (**C**,**D**) the landscape of the PPI network and the significant subnetworks (mcodes) originating from the overall PPI network of the MCF-7 and MDA-MB-231 cells.

**Figure 4 ijms-24-12479-f004:**
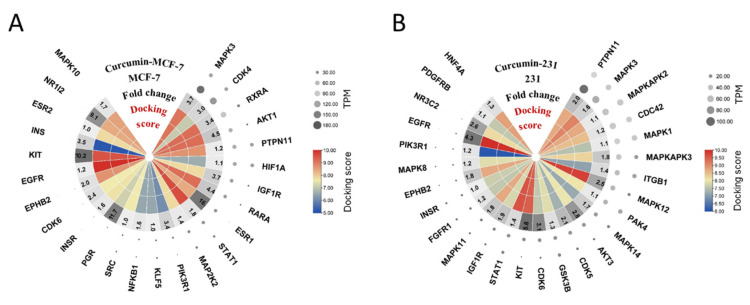
Screening and docking evaluation of effective superimposed attenuation targets. (**A**,**B**) Heatmaps of superimposed attenuation targets of curcumin in MCF-7 and MDA -MB-231 cell lines; (**C**) effective superimposed attenuation targets of curcumin were used to map the docking score (red for high and blue for low) to the breast cancer KEGG pathway map; (**D**–**K**) docking poses and 2D ligand–protein interaction of candidate targets with curcumin and inhibitor docked on same cavity provided by the template.

**Figure 5 ijms-24-12479-f005:**
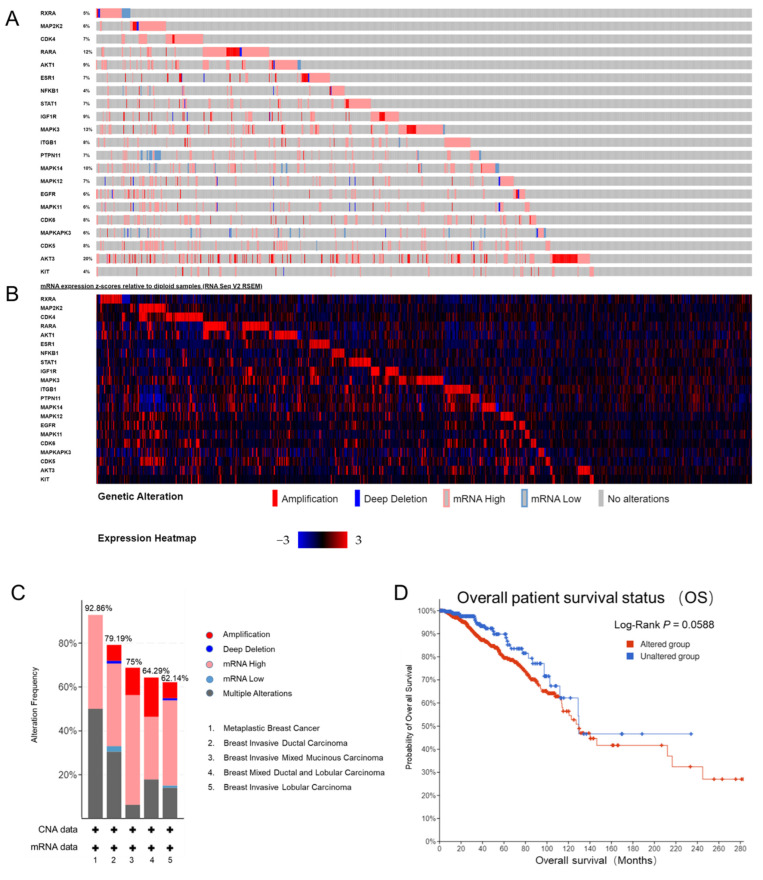
Genetic alteration and prognostic value of curcumin. (**A**,**B**) OncoPrint visual summary of 21 gene alterations and mRNA expression in the breast invasive dataset (TCGA, Firehose Legacy) in cBioPortal; (**C**) summary of alterations in 21 curcumin targets in 5 breast cancer types (TCGA, Firehose Legacy); (**D**) OS analysis in cases with or without alterations in curcumin targets of the TCGA dataset; (**E**) the relationships between RFS and the expression of 5 curcumin targets in TNBC patients.

**Figure 6 ijms-24-12479-f006:**
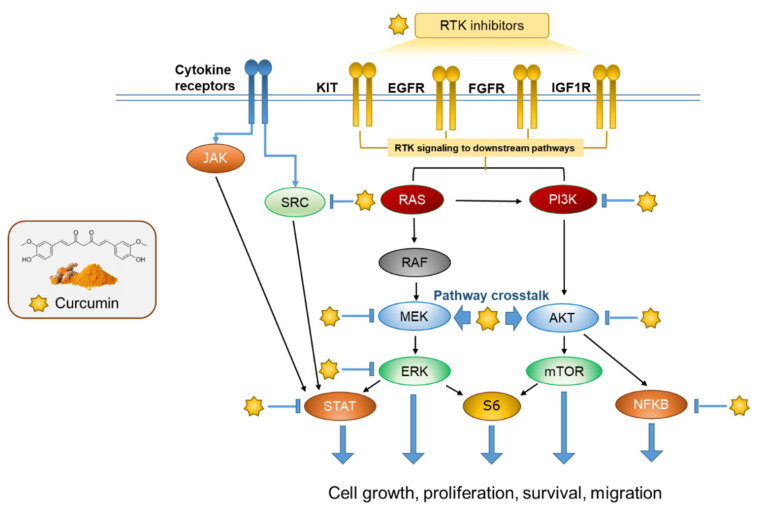
Curcumin inhibits the progression of breast cancer by regulating superimposed attenuation targets in multiple signal transduction pathways.

## Data Availability

All sequence reads that were used in this study can be found under the BioProject PRJNA613560.

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
