# Peer review of "Integrated Transcriptome and Molecular Docking to Identify the Hub Superimposed Attenuation Targets of Curcumin in Breast Cancer Cells"

_ijms, 2023, doi:10.3390/ijms241512479_

Round 1
Reviewer 1 Report
In manuscript entitled “ Integrated transcriptome and molecular docking to identify the hub superimposed attenuation targets of curcumin in breast cancer cells” authors employed RNA-Seq and various in-silico methodologies to investigate the possible anti-cancer mechanisms associated with curcumin effect in breast cancer. Their study utilized the MCF-7 and MDA 231 cell lines as representative models. Overall, the article is well-written and presents information in a straightforward manner., however I have following comments/suggestions
It is well-known that curcumin exhibits substantial anti-cancer effects, as supported by numerous preclinical studies. However, concerns regarding the efficacy and safety of curcumin have persisted. In this study, the authors investigated the impact of curcumin on two breast cell lines. To enhance the comprehensiveness of their analysis, I recommend that the authors include additional normal cell lines to assess the effects of curcumin and evaluate its potential toxicity.
Furthermore, considering that the two breast cancer cell lines used in this study exhibit distinct expression profiles, it may be insufficient to solely rely on these cell lines to support the authors' conclusions. I also suggest authors to use at least two cell lines with similar expression profiles to confirm their initial results.
Also, considering the authors' exploration of various targets associated with the anticancer effects of curcumin, it would be intriguing to investigate the expression profiles of these targets in breast cancer patients. To achieve this, the authors can potentially use publicly available expression data from breast cancer patients from existing databases. This additional analysis would further enhance the significance and translational potential of the authors' findings.
In lines 195-197, the authors state that their study is the first to utilize transcriptomic techniques to evaluate transcriptional changes in the downstream targets of curcumin, leading to the identification of attenuated targets with superimposed effects. However, this claim may not hold true as several studies in recent years have employed similar techniques to investigate similar aspects. For instance, numerous research studies have already been conducted using comparable approaches. It is suggested that the authors revise this statement to accurately reflect the novelty and contribution of their study in the context of existing literature
Nirgude, Snehal, Sagar Desai, and Bibha Choudhary. "Curcumin alters distinct molecular pathways in breast cancer subtypes revealed by integrated miRNA/mRNA expression analysis." Cancer Reports 5.10 (2022): e1596.
Wang, Rong, et al. "Investigating the therapeutic potential and mechanism of curcumin in breast cancer based on RNA sequencing and bioinformatics analysis." Breast Cancer 25 (2018): 206-212
Author Response
Q1. It is well-known that curcumin exhibits substantial anti-cancer effects, as supported by numerous preclinical studies. However, concerns regarding the efficacy and safety of curcumin have persisted. In this study, the authors investigated the impact of curcumin on two breast cell lines. To enhance the comprehensiveness of their analysis, I recommend that the authors include additional normal cell lines to assess the effects of curcumin and evaluate its potential toxicity.
Response:Thank you very much for your request for additional tests for the cytotoxicity of curcumin, we have completed the cell viability assay of curcumin in MCF-10A (human normal mammary epithelial cells), and according to your request, we have added the text description in lines 100-102, and the results of MCF-10A cell viability assay is shown in Figure S1.
Q2. Furthermore, considering that the two breast cancer cell lines used in this study exhibit distinct expression profiles, it may be insufficient to solely rely on these cell lines to support the authors' conclusions. I also suggest authors to use at least two cell lines with similar expression profiles to confirm their initial results.
Response:Thank you very much for your careful observation, and please forgive us for the unclear description of the criteria for cell sample selection. The selection of two distinct cell lines was our initial intention to demonstrate that curcumin can inhibit the proliferation and development of various breast cancer cells by targeting different mechanisms. According to the principle of estrogen receptor typing, breast cancer cell lines can be categorized as ER-positive and ER-negative. MCF-7 (ER+) and MDA-MB-231 (ER-) are the most commonly used representative breast cancer cell lines. The wide application of MCF-7 and MDA-MB-231 in recent decades confirms that they can be used for screening of most breast cancer drugs and studying pathogenesis. We have added the description of the experimental purpose in line 16 and lines 223-232. We hope that the above revisions will receive your understanding and support.
Q3. Also, considering the authors' exploration of various targets associated with the anticancer effects of curcumin, it would be intriguing to investigate the expression profiles of these targets in breast cancer patients. To achieve this, the authors can potentially use publicly available expression data from breast cancer patients from existing databases. This additional analysis would further enhance the significance and translational potential of the authors' findings.
Response:Thank you very much for your valuable suggestion, we added a new result “2.5. Genetic alteration and prognostic value of curcumin targets in patients with breast cancer” specifically as you requested to elucidate the genetic alterations and mRNA expression of curcumin's targets in samples from breast cancer patients with TCGA and assessed their impact on prognosis by survival analysis in lines 181-207.
Q4. In lines 195-197, the authors state that their study is the first to utilize transcriptomic techniques to evaluate transcriptional changes in the downstream targets of curcumin, leading to the identification of attenuated targets with superimposed effects. However, this claim may not hold true as several studies in recent years have employed similar techniques to investigate similar aspects. For instance, numerous research studies have already been conducted using comparable approaches. It is suggested that the authors revise this statement to accurately reflect the novelty and contribution of their study in the context of existing literature.
Response:Thank you very much for the correction on the write-up, we have removed this inappropriate description as per your request.
Reviewer 2 Report
Wang et al. intended to discuss the integration of transcriptomic and molecular docking approaches which can identify the inhibitory targets of curcumin in breast cancer, in their research article " Integrated transcriptome and molecular docking to identify the hub superimposed attenuation targets of curcumin in breast cancer cells."
This constitutes a comprehensive, and interesting topic of research work that is appreciated. However, there are minor concerns that can be resolved to improve the quality of the manuscript, listed below.
1. The utilization of curcumin in the treatment of cancer, specifically breast cancer, is not a novel concept. Therefore, it would be more pertinent to highlight the significance of high-throughput screening analysis rather than solely emphasizing the importance of curcumin as an approach to therapy.
2. The introduction should provide a more comprehensive discussion on how your study is contributing new significance to the existing body of research.
3. Please provide a brief analysis of the clinical significance of this study.
4. There is space for grammatical improvement.
There is space for grammatical improvement and sentence reconstruction for better understanding.
Author Response
Reviewer #2:
Q1. The utilization of curcumin in the treatment of cancer, specifically breast cancer, is not a novel concept. Therefore, it would be more pertinent to highlight the significance of high-throughput screening analysis rather than solely emphasizing the importance of curcumin as an approach to therapy.
Response:Thank you very much for your attention to the highlights of the article. We have revised the purpose of this study in the introduction (lines 69-71) and conclusion (lines 299-305). I hope the changes mentioned above will meet your satisfaction.
Q2. The introduction should provide a more comprehensive discussion on how your study is contributing new significance to the existing body of research.
Response:According to your guidance, we have rewritten the first paragraph (lines 209-222) of the discussion to provide a comprehensive explanation of the new significance of this study.
Q3. Please provide a brief analysis of the clinical significance of this study.
Response:Based on your valuable suggestion, we have added a new independent analysis “2.5. Genetic alteration and prognostic value of curcumin targets in patients with breast cancer” investigates the genetic variation and prognostic analysis of curcumin targets. This study aims to identify potential targets for curcumin in individualized medicine. In lines 181-207.
Q4. There is space for grammatical improvement.
Response:Thank you very much for your attention to grammar. Due to the limited time for revision, we have to proofread the grammar of the manuscript ourselves. I hope our efforts will be recognized by you.
Round 2
Reviewer 1 Report
The authors have diligently addressed all my concerns and carefully incorporated the changes suggested during the review process. Their efforts have significantly improved the quality and rigor of the research article, making it now suitable for publication.